



# Airborne in-situ quantification of methane emissions from oil and gas production in Romania

Hossein Maazallahi[1,*], Foteini Stavropoulou[1,6], Samuel Jonson Sutanto[1,**], Michael Steiner[2], Dominik Brunner[2,3], Mariano Mertens[4], Patrick Jöckel[4], Antoon Visschedijk[5], Hugo Denier van der Gon[5], Stijn Dellaert[5], Nataly Velandia Salinas[6], Stefan Schwietzke[6], Daniel Zavala Araiza[6], Sorin Ghemulet[7], Alexandru Pana[7], Magdalena Ardelean[7], Marius Corbu[7], Andreea Calcan[7], Stephen A. Conley[8], Mackenzie L. Smith[8], Thomas Röckmann[1]

[1] Institute for Marine and Atmospheric research Utrecht (IMAU), Utrecht University, Utrecht, the Netherlands

[2] Laboratory for Air Pollution/Environmental Technology, Empa - Swiss Federal Laboratories for Materials Science and Technology, Dübendorf, Switzerland

[3] Institute for Atmospheric and Climate Science, ETH Zurich, Zurich, Switzerland

[4] Deutsches Zentrum für Luft- und Raumfahrt, Institut für Physik der Atmosphäre, Oberpfaffenhofen, Germany

[5] Netherlands Organisation for Applied Scientific Research (TNO), Utrecht, the Netherlands

[6] Environmental Defense Fund (EDF), Berlin, Germany, and Amsterdam, the Netherlands

[7] National Institute for Aerospace Research "Elie Carafoli" (INCAS), Bucharest, Romania

[8] Scientific Aviation (SA), Inc., 3335 Airport Road Suite B, Boulder, Colorado 80301, United States

[*] Now at: Department of Renewable Energies and Environment, College of Interdisciplinary Science and Technologies, University of Tehran, Tehran, Iran.

[**] Now at: Earth Systems and Global Change, Wageningen University and Research, Wageningen, the Netherlands.

**Correspondence to**:

Hossein Maazallahi (h.maazallahi@ut.ac.ir), Thomas Röckmann (t.roeckmann@uu.nl)

**Abstract**

Production of oil and gas in Romania, one of the largest producers in the EU, is associated with substantial emissions of methane to the atmosphere and may offer high emission mitigation potential to reach the climate objectives of the EU. However, comprehensive quantification of emissions in this area has been lacking. Here we report top-down emission rate estimates derived from aircraft-based in-situ measurements that were carried out with two aircraft during the ROMEO 2019 campaign, supported by simulations with atmospheric models. Estimates from mass balance flights at individual dense production clusters, and around larger regions, show large variations between the clusters, supporting the important role of individual super emitters, and possibly variable operation practices or maintenance state across the production basin. Estimated annual total emissions from the Southern Romanian O&G infrastructure are $227 \pm 86$ kt $CH_4$ yr$^{-1}$, consistent with previously published estimates from ground-based site-level measurements during the same period. The comparison of individual plumes between measurements and atmospheric model simulations was complicated by unfavorable low wind





conditions. Similar correlations between measured and simulated $CH_4$ enhancements during
large-scale raster flights and mass balance flights suggest that the emission factor determined
from a limited number of production clusters is representative for the larger regions. We
conclude that ground-based and aerial emission rate estimates derived from the ROMEO
campaign agree well, and the aircraft observations support the previously suggested large
under-reporting of $CH_4$ emissions from the Romanian O&G industry in 2019 to UNFCCC.

## 1. Introduction

Methane ($CH_4$) is a potent greenhouse gas with more than 80 times the global warming
potential of carbon dioxide ($CO_2$) over a 20-year time horizon (Szopa et al., 2021).
Approximately 60% of global $CH_4$ emissions are attributed to human activities, with roughly
one-third of them resulting from the Oil and Gas (O&G) industry (Saunois et al., 2020).
Reducing $CH_4$ emissions from the O&G industry presents an easily accessible and cost-
effective mitigation option (Shindell et al., 2021). Given the relatively short lifetime of $CH_4$ in
the atmosphere ($\approx 10$ years), such measures would lead to substantial climate benefits in both
the near- and long-term future (Shindell et al., 2021;Collins et al., 2018). Scenarios that are
compatible with the goal of the Paris Agreement (UNFCCC, 2015) to limit global warming to
2 °C, preferentially to 1.5 °C all include substation reductions in $CH_4$, and the current growth
in $CH_4$ is incompatible with reaching this goal (Nisbet et al., 2020).
Improving our understanding of $CH_4$ emissions from the O&G industry requires
comprehensive and accurate emissions measurements using a combination of approaches.
Several studies, mostly in North America, consistently show that national inventories, which
rely on multiplying activity data with generic emission factors, tend to underestimate $CH_4$
emissions from the O&G industry (Allen et al., 2013;Brandt et al., 2014;Harriss et al.,
2015;Johnson et al., 2017;Alvarez et al., 2018;Weller et al., 2020).
$CH_4$ emissions can be quantified using top-down or bottom-up approaches. Top-down
approaches use ambient $CH_4$ mole fraction measurements from aircraft, tall towers, weather
stations or satellites, combined with models to estimate the total $CH_4$ flux rate at different scales
(i.e., site-level to regional or country-level). These approaches ensure that emissions from all
sources are captured. Other techniques, such as the use of ethane ($C_2H_6$) and the isotopic
composition of $CH_4$ as tracers, can help attribute $CH_4$ emissions to O&G industry or other
sectors (Röckmann et al., 2016;Lopez et al., 2017;Mielke-Maday et al., 2019;Maazallahi et al.,
2020;Lu et al., 2021;Menoud et al., 2021;Gonzalez Moguel et al., 2022;Fernandez et al., 2022).
Bottom-up approaches involve direct measurements of emissions usually at the source or
component-level which are then extrapolated to larger scales using statistical methods.
The emission data reported to the United Nations Framework Convention on Climate
Change (UNFCCC) for the year 2021 reveal that Romania ranks among the European Union
(EU) countries with the highest annual $CH_4$ emissions from the O&G activities, following
closely behind Italy and Poland. The International Energy Agency (IEA) estimates that
Romania contributes the highest $CH_4$ emissions from the O&G industry among the EU-27
countries (IEA, 2023). In light of the recent provisional agreement of EU methane regulations,
which impose new requirements on the O&G industry for measuring, reporting, and mitigating
$CH_4$ emissions (European-Commission, 2023), there is an urgent need to understand the extent
and magnitude of emissions. This is particularly relevant for countries like Romania, where
emissions are substantial but understudied, and addressing them is crucial for achieving EU
climate objectives.
The ROMEO (ROmanian Methane Emissions from Oil and gas) project was designed
to provide independent scientific measurement based $CH_4$ emission estimates for the O&G
producing regions in Romania (Stavropoulou et al., 2023). The first phase of the ROMEO



campaign took place in October 2019, covering large production areas in southern Romania
that are mostly associated with oil production. Numerous measurement techniques using a
variety of instruments were deployed onboard ground-based and airborne measurement
platforms. The data collected by vehicles and UAVs during the ROMEO campaign have
already been evaluated separately in prior studies (Stavropoulou et al., 2023;Delre et al.,
2022;Korbeń et al., 2022). Additionally, Menoud et al. (2022) investigated the isotopic
signature of $CH_4$ emissions from the sites visited during the ROMEO campaign, contributing
to insights in the reservoir characteristics.
In this study, we present top-down $CH_4$ emission estimates derived from aircraft
measurements of individual facilities, facility clusters, and extended regions during the
ROMEO campaign. The measurements were performed by two research aircraft, and we used
two mesoscale atmospheric chemistry and transport models to simulate atmospheric
composition and transport over Romania.

## 2. Materials and methods
**2. Materials and methods**
**2.1. Clusters and regions**
Information of O&G activities including locations, productions asset types, status and
age of the facilities were received from the largest operator in the region. This information
covers the majority of the total sites in the survey region, where other smaller operators are
also present. The distribution of O&G production infrastructure in Romania is heterogeneous
with a high density of production sites concentrated above the subsurface fossil fuel reservoirs.
Therefore, we first grouped the installations in 40 clusters (Cs) and regions (Rs) (i.e.,
aggregation of several production clusters). Both production clusters and regions were targets
for the quantification approaches in the ROMEO campaign. Clusters are relatively small areas,
usually a few to 20 $km^2$, with a high density of O&G production sites. To derive basin-scale
emission rates from aircraft measurements, the Romanian plain was further divided into larger
regions of roughly 50 x 50 $km^2$, which contain the clusters and are suitable for aircraft mass
balance and raster flights.

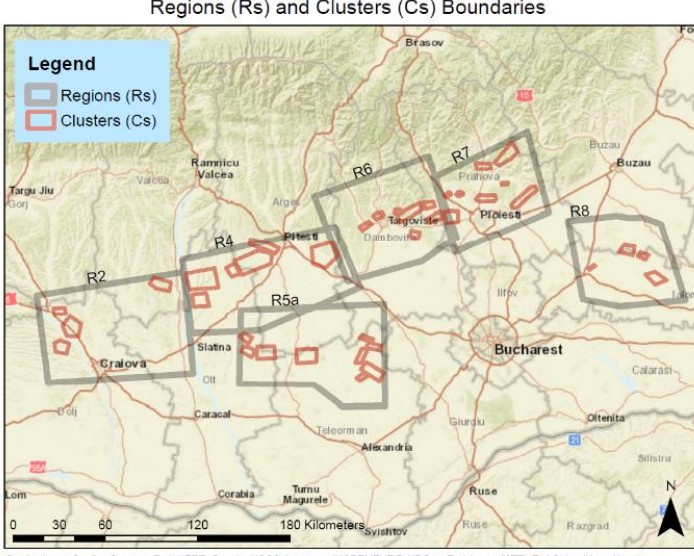






*Figure 1. Regions (grey polygons) and clusters (red polygons) that were targeted during the*
*ROMEO 2019 campaign, circular or raster flights were performed within or around these*
*boundaries. Black symbols are individual sites of the O&G production infrastructure.*

### 2.2. Aircraft-based in situ measurements

Two aircraft were deployed during the ROMEO 2019 campaign, a BN2 aircraft
operated by the National Institute for Aerospace Research "Elie Carafoli" (INCAS) and a two-
seater Mooney aircraft operated by Scientific Aviation (SA) Inc. On the Mooney aircraft, in-
situ measurements of $CH_4$, $C_2H_6$, carbon dioxide ($CO_2$), wind speed and direction, and relative
humidity were continuously logged at 1Hz frequency. $C_2H_6$ and $CO_2$ were measured with
AERIS Pico Mobile LDS and Picarro G2301-f instruments and both instruments measured $CH_4$
individually. On the BN2 aircraft, $CH_4$, $CO_2$ and carbon monoxide (CO) were measured at
about 0.3 Hz frequency using a G2401 analyzer (Picarro Inc).
Two sets of flight patterns were performed, mass balance flights circling around target
areas (Fig. 2, left) and raster flights scanning the areas at a pre-selected observation density
(Fig. 2, right). During the 18 individual mass balance flights with the SA aircraft, the target
emission locations were circled at different altitudes to map the extent of the emission plume
(s), both vertically and horizontally. The emission rates were then calculated from the
measurements of $CH_4$ mole fraction and wind speed and direction in the mass balance approach
(see below). The BN2 aircraft was used to map possible emission sources over more extended
areas. The lack of wind measurements from this aircraft precludes a direct emission
quantification using the mass balance approach. These extended areas were surveyed in raster
patterns perpendicular to the prevailing wind (Fig. 2b). In addition to the identification of larger
sources, these measurements are also used to derive indirect emission rate estimates by
comparison to model simulations (see below).

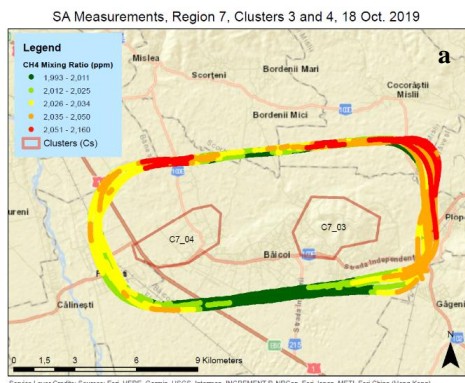
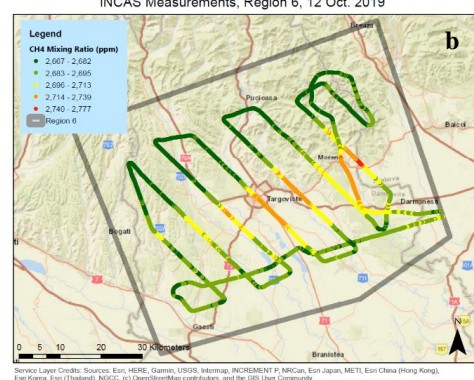

*Figure 2 – Examples of a mass balance flight (a) and a raster flight (b) during the ROMEO*
*2019 campaign. Black symbols are individual sites of the O&G production infrastructure. The*
*mass balance flight circled around two production clusters located in close proximity and the*
*raster flight covers a larger region. The color scale represents the $CH_4$ mole fraction.*

### 2.3. Model simulations

In order to support the emission quantification from the aircraft measurements, we
simulated atmospheric composition and transport over Romania using two numerical mesoscale
atmospheric chemistry and transport models: COSMO-GHG operated by the Swiss Federal
Laboratories for Materials Science and Technology (EMPA) and MECO(n) operated by the



German Aerospace Center (DLR). COSMO-GHG is based on the regional numerical weather prediction and climate model COSMO-CLM (Baldauf et al., 2011) and includes the GHG extension (Jähn et al., 2020;Brunner et al., 2019) for the simulation of (nearly) passive trace gases such as $CH_4$. MECO(n) features an on-line coupling of the global chemistry-climate model EMAC with the regional chemistry-climate model COSMO-CLM/MESSy (Kerkweg and Jöckel, 2012). The COSMO-GHG simulations were nudged to the hourly wind data from the ERA5 reanalysis product of the European Centre for Medium-Range Weather Forecasts (ECMWF) (Hersbach et al., 2023). In MECO(n) the global model (EMAC) was nudged by Newtonian relaxation towards the operational analysis data from ECMWF (see (Nickl et al., 2020) for more details).

These two models were used to simulate the evolution of the $CH_4$ mole fraction arising from emissions from active O&G assets, including individual wells and larger facilities in time and space. For setting up the model simulations, each site was assigned an emission rate of 1 g $s^{-1}$ (3.6 kg $hr^{-1}$). For COSMO-GHG, the model resolution was 2 x 2 $km^2$, and the meteorological and compositional boundary conditions were provided from global scale modeling results obtained with the ECMWF/CAMS system. The MECO(n=3) set-up comprised four model instances (see (Klausner et al., 2020) for a detailed description of a similar model set-up). The first is the global model instance EMAC with a resolution of T42L90MA (corresponding to around 280 km spatial resolution). In the global model, three COSMO-CLM/MESSy instances were nested on-line with approx. 50 km resolution, approx. 7 km resolution, and the same 2 x 2 km domain as applied for COSMO-GHG, respectively. In the applied MECO(3) set-up, we used a parameterized chemistry of methane (Winterstein and Jöckel, 2021)with monthly mean OH fields from previous simulations with comprehensive interactive chemistry. In the first, second, and third MECO(3) model instance we prescribed all anthropogenic and natural emissions of methane, in order to achieve realistic boundary conditions of methane for the finest resolved instance. In this instance the emissions were used as described below. The model outputs provide atmospheric $CH_4$ mole fractions fields as well as meteorological parameters at a temporal resolution of 20 min (COSMO-GHG) and 1 hr (MECO(3)). For MECO(3), only the results of the finest instance are considered here for further analysis. To be able to attribute emissions to certain emission clusters, we transported 33 individual "$CH_4$ tracers" based on 3 inventories. 21 of these tracers represent the emissions of individual clusters and regions, and in some cases even represent gas and oil emissions separately. To limit the number of tracers, one tracer represents the emissions of one or two clusters and one or two distant regions, assuming that they are sufficiently far away. This allows us to separate the signal of each cluster flown over or circled around.

## 2.4. Emission inventories

To drive the simulations and interpret the data we use information from various emission inventories. (1) The most granular dataset is based on information on the production infrastructure provided by the oil and gas operator. It consists of about 6000 individual production-related locations in the Southern part of Romania. We will refer to this dataset as the "O&G_operator" dataset. In order to convert this to an approximate emission inventory, we divided reported emissions for Romania by the number of emission locations and assigned the result as average emission rate to all of these locations. Coincidentally, this average value is close to 1 g $s^{-1}$ $site^{-1}$ (3.6 kg $hr^{-1}$ $site^{-1}$), which was used as prior emission rate in the model simulations. (2) The TNO_GHGco inventory (Denier van der Gon et al., 2018) includes emissions from all available sectors at 5 km x 5 km resolution. (3) The European Pollutant Release and Transfer Register/Industrial Emissions Directive (E-PRTR/IED) inventory (E-PRTR, 2023) includes major point sources and was used to identify major farm and landfill methane emitters within the study areas (Figure S2 in the SI). (4) The TNO - Copernicus



Atmospheric Monitoring Service European Regional Inventory (TNO-CAMS) (Kuenen et al.,
2022) and (5) the Emissions Database for Global Atmospheric Research (EDGAR, 2023)
inventories were used to calculate the percentage of O&G emissions to total emissions in the
target areas.
In summary, based on TNO-CAMS no coal mine locations, a potentially large source
of $CH_4$, were identified within the mass balance flight boundaries. The presence of major
wetlands was investigated based on the findings of Saarnio et al. (2009) and no wetlands were
observed within the measured areas.

### 2.5. Analysis of simulated meteorological quantities

The meteorological conditions during the ROMEO campaign were not ideal for
emission quantification due to the low wind speeds. This complicated the use of a model –
measurement comparison for the raster flights, which we had planned to use to derive
quantitative emission information. To assess the model performance in terms of meteorological
conditions during the individual flight days, we compared the meteorological output of the
models with each other, with ERA5 reanalysis data, and with the meteorological information
recorded during the Scientific Aviation flights. The rationale is: when the models do not agree
on the general meteorological conditions in a target region, we also expect diverging $CH_4$
concentration distributions, which would hamper quantitative comparison to the measurements.
On the other hand, when the meteorological conditions are simulated consistently, there is more
confidence that the transport is simulated adequately as well, thus the simulated and observed
$CH_4$ plumes may be used to derive emission information.
For each flight date, the following parameters were investigated in each flight region:
temperature, cloud fraction, wind speed and direction, specific humidity, and relative humidity.
Based on selected threshold values, the meteorological parameters for each model and each
flight day were characterized as good, acceptable, or poor. Furthermore, we evaluate three
quantitative indices, the Nash - Sutcliffe Efficiency (NSE), the Kling-Gupta Efficiency (KGE),
and the Mean Absolute Relative Error (MARE) between simulation results and ERA5
reanalysis data. The results of this comprehensive analysis are presented in the Supplementary
Information (SI).

### 2.6. Emission quantification: Mass balance approach

$CH_4$ emission rates from 11 production site clusters (or combinations of clusters), three
larger regions in the Romanian Basin, and two groups of individual sites were quantified from
aircraft-based measurements using the mass balance approach. This approach is based on the
Gaussian theorem in which the difference of the total fluxes into and out of an enclosed area
must be balanced by a source or sink in the area (Conley et al., 2017). $CH_4$ enhancements were
identified using background values determined either from the upwind flight legs or from the
edges of detected plumes.
The mass balance approach returns total $CH_4$ emissions for the target areas. For the
intense production clusters, the emissions are in most cases dominated by the O&G production
infrastructure. Therefore, we assigned 100% of the emissions in the clusters to O&G
production, except for clusters which contained a landfill and/or large farm, as included in the
E-PRTR inventory. In particular, only one significant landfill was identified in R6C6, and the
emissions reported from this landfill were deducted from the measured flight quantification.
For the larger areas, the contributions from other sectors can be substantial. To infer emissions
related to O&G operations from the total measured emissions, we estimated the emissions from
non-O&G sources in the target areas using the TNO-CAMS inventory and subtracted these
from the total measured emissions. We repeated the same process using the EDGAR inventory.



These O&G related emissions were then divided by the number of active O&G infrastructure
elements in the target area to derive an emission factor per site for that cluster or region. This
includes active production sites, processing sites, compressor stations, and other active sites,
which all contribute to the measured emissions. Possible emissions of non-producing sites are
not included in our estimates, as they are likely smaller (on average) than the ones of producing
sites.

### 2.7. Emission quantification: Measurement - model comparison
#### 2.7.1. Mass balance flights

The simulated $CH_4$ distributions were evaluated along the flight tracks in order to
facilitate direct comparison with the observations. For the mass balance flights (Fig 2a), the
lowest $CH_4$ value of each circle around a target area was defined as background mole fraction
and subtracted from downwind measurements to obtain the $CH_4$ enhancement. To compare
model and measurement results, we integrated the $CH_4$ enhancement above background along
the flight track for each circle, both for the measurements and for the simulated $CH_4$ mole
fractions along the flight tracks. These integrals are referred to as plume areas. Circles that were
identified as influenced by up-stream contamination were excluded from the analysis. The
simulated plume areas were then plotted versus the measured plume areas, and the slope of the
orthogonal linear regression line returns a measurement-based scaling factor to the prior
emission rate estimate that was in the simulations (1 g s$^{-1}$ site$^{-1}$). This scaling factor was then
assigned to the active O&G facilities in the target cluster or region and provides a measurement-
based estimate of the emission factor.

#### 2.7.2. Raster flights

For the raster flights (Fig. 2b), the lowest $CH_4$ mole fraction along the flight track across
a target region was defined as background and the $CH_4$ enhancements above this background
were integrated. The simulations were treated in the same way. The slopes of the orthogonal
linear regressions between integrated enhancements from flight measurements and simulations
were then compared to the scaling factors determined from the mass balance flights (2.7.1) to
investigate whether the model – observation slopes are consistent between individual plumes
and the raster flights over larger regions. The rationale is that even if the quantitative modeling
is challenging under the encountered meteorological conditions, if the slopes derived from the
mass balance and raster flights are comparable, then the emission factors derived from the mass
balance flights should be also representative for the larger regions covered by the raster flights.

### 3. Results and discussion

### 3.1. Mass balance quantifications

Table 1 shows the results of the emissions quantifications obtained from mass balance
calculations using the measurements of the SA aircraft. Methane emission rates range between
tens of kg hr$^{-1}$ from an individual facility or smaller cluster up to more than 8000 kg hr$^{-1}$ for the
larger region R7 which includes the city of Ploiesti. These emissions are representative of the
sum of all sources in each target area. Especially the larger regions include emissions from other
sectors, particularly agriculture and waste. On the other hand, the $CH_4$ in the dense production
clusters originate to nearly 100% from O&G activities.
Different inventories (E-PRTR, TNO-CAMS and EDGAR) were consulted to obtain
information about the non-O&G contributions; however, these inventories are generally not
designed to distribute emissions across sectors on such small scales. TNO-CAMS and EDGAR
have a coarse spatial resolution and do not include production clusters, so they are not suitable
to assess the emissions distribution across sectors in such clusters. With the exception of R6C6,



which includes a landfill listed in E-PRTR, for all other production clusters, E-PRTR does not
indicate any major farms or landfills. The ground teams did not observe significant non-O&G
sources in the smaller production clusters. Therefore, we ascribe 100% of the total emissions in
clusters to O&G production. For the large regions R7 and R5a, we use the estimated absolute
non-O&G emissions from TNO-CAMS and subtract them from the measured emissions to
correct for non-O&G related emissions.
The emission factors (EF) provided in Table 1 are calculated using the number of total
active (e.g. producing, or operating) infrastructure elements within the target regions, because
the measurements do not allow us to distinguish between different parts of the infrastructure.
The emission factors vary widely among the individual clusters, from 1.0 to 20 kg hr$^{-1}$ site$^{-1}$.
This is partly due to the inhomogeneous distribution of the emissions, where few sites are
responsible for a large share of the emissions.  A contributing factor is that each quantification
yields an emission estimate for the specific moment in time of the measurement. The variability
in our cluster-specific emission factors may partly represent the episodic tendency of O&G
super-emitters. However, given the generally large number of infrastructure elements within
the target regions, the reported numbers should still reflect representative averages for the
clusters and regions, also over longer periods. Note that the timing of our measurements is
random, and the total facility sample size (N=4358, including duplicates, see below) is large.
To address the challenge of emissions' variability and inhomogeneity, we employ a weighted
averaging approach based on facility numbers.
Table 1 shows that over all mass balance flights around production clusters and larger
regions, a total of roughly 31,700 kg hr$^{-1}$ of $CH_4$ emissions were quantified. Dividing this by
the number of active facilities in the target regions (4358) would lead to an emission factor of
7.3 kg hr$^{-1}$ site$^{-1}$. However, this number is biased high for two reasons: i) not all of the measured
emissions originate from the O&G production facilities, in particular for the large regions, and
ii) there is some "double" or even "triple counting" of sites. Specifically, the emission from the
larger regions R5a and R7 were quantified twice, and the sites in the quantified clusters of R7
contribute to the emissions from the clusters and the regional flights. Note that the activity
factors are also counted several two or three times in this case, so the calculations are still valid,
but sites that are quantified multiple times have more weight in calculation of the EF.
To exclude double or triple counting emissions from the same facilities, we estimate an
EF based solely on the larger region quantifications for R5a and R7. In the case of regions R4,
R8 and R6 where no regional quantification was performed or regional quantifications were
unsuccessful, we aggregate the emissions from the individual production clusters. When we
remove the double counting and account for emissions from other sources as explained above,
the total emissions quantified over these mass balance flights are 13,216 ± 4932 kg hr$^{-1}$, which
results in a facility-weighted emission factor of 5.3 ± 2.0 kg hr$^{-1}$ site$^{-1}$. We note that these
estimates from the mass balance flights represent a large fraction of the O&G infrastructure in
the Southern Romanian region (2516 total sites, Table 1) and are thus statistically representative
of the area.
By design, regions with more sites contribute more to facility-weighted emission factors
than regions with less sites. In our case, region R5a has nearly twice as many facilities as the
next most densely populated region and thus carries the largest weight in the calculations. We
recall that the weather conditions during the ROMEO campaign were not ideal for large scale
mass balance quantifications. It is possible that the area mass balance quantifications in the flat
and arid region R5a in Southern Romania may be biased slightly low due to partial loss of $CH_4$
out of the boundary layer during the hot and convective conditions, or due to the fact that stable
transport conditions had not yet established over the large regions (e.g. not all emissions in the



region had time to leave through the downwind edge where the measurements took place). In addition, the results from the region R5a flights were affected by upwind emissions.

For flights around dense production clusters with 100% O&G contribution, the weighted average emissions factor is $4.4 \pm 1.7$ kg hr$^{-1}$ site$^{-1}$. This aligns with the estimate from the large regions within the uncertainty ranges (EF of $5.3 \pm 2.0$ kg hr$^{-1}$ site$^{-1}$, Table 1), considering the differences in the number of facilities included in each estimate (N=1570 for the clusters and N=2516 for the large regions). These estimates also agree within the errors with the 5.4 (3.6 – 8.4) kg hr$^{-1}$ oil production site$^{-1}$ (95% CI: 3.6 – 8.4) reported from the ground-based measurements by Stavropoulou et al. (2023) for oil production sites. The estimate derived from the cluster flights encompasses a total of 1570 facilities, compared to up-scaling from about dedicated measurements at 178 oil production sites in Stavropoulou et al. (2023). The emissions quantified with the aircraft include all facilities in a certain region or cluster, not only the production facilities. Other facilities include facilities that may emit more than an individual well (e.g. oil parks) but also facilities that emit less (e.g. injection or disposal facilities). If the co-produced gas from oil production sites is already vented at the production sites as reported in (Stavropoulou et al., 2023), then the production sites may indeed be larger emitters than the average, because most of the gas already escapes at the production site.

As the campaign airport was located close to the city of Ploiesti in region R7, the majority of cluster quantifications were carried out in R7 for logistical reasons and many of the dense production clusters in R7 were quantified. This allows us to compare the sum of the emission rates determined from cluster quantifications to the emission factors from regional quantifications. The cluster flights in region R7 quantified a total of 377 O&G sites, which is 75% of the 500 sites that were quantified in the regional flights. The quantified emissions from the cluster flights ($3828 \pm 1199$ kg hr$^{-1}$) amount to 54% of the total emissions quantified in the regional flights, after subtracting non-O&G emissions (about $7038 \pm 1769$ kg hr$^{-1}$ from two independent flights, Table 1). This indicates a possible underestimate of non-O&G emissions in the inventories for R7, which includes the large city of Ploiesti. Alternatively, some super-emitters may exist outside the quantified clusters, which would increase the regional estimate. Nevertheless, the region and cluster slights show a reasonable level of consistency in region R7. The emission factors further support this alignment, with the weighted sum of the clusters being equal to $10.2 \pm 3.2$ kg hr$^{-1}$ site$^{-1}$ compared to about $14.1 \pm 3.6$ kg hr$^{-1}$ site$^{-1}$ for the regional flights.

The aircraft-based quantifications indicate that per-site emission factors from region R7 are higher than from the other regions. At the same time, R7 was best covered in terms of mass balance determinations, so it is the most reliable estimate. From the site-level quantifications carried out on the ground, it was not apparent that per-site emission rates varied between different regions (Stavropoulou et al., 2023;Delre et al., 2022;Korbeń et al., 2022).

When we use the derived emission factor of $5.3 \pm 2.0$ kg hr$^{-1}$ site$^{-1}$ and scale this up to the entire production basin in Southern Romania with more than 4900 active sites, annual estimated emissions estimated at $227 \pm 86$ kt $CH_4$ yr$^{-1}$. If the derived EF also applies to the infrastructure in other parts of Romania the inferred country-scale emission rate from about 7400 active sites in 2019 is $344 \pm 130$ kt $CH_4$ yr$^{-1}$. Reported emissions to the UNFCCC for Romania in the category *1.B: Fugitives* include 53 kt $CH_4$ yr$^{-1}$ for activity *1.B.2.b Natural Gas*, 38.2 kt $CH_4$ yr$^{-1}$ for *1.B.2.c Venting and Flaring (oil, gas, combined oil and gas)* and 10.4 kt yr$^{-1}$ for *1.B.2.a Oil* (UNFCCC, 2023b). This adds up to 101.6 kt $CH_4$ yr$^{-1}$, about 3 times less than our estimate. Our estimate does not include emissions from infrastructure operated by other operators, for example the large gas production region in the Transylvanian Basin. An intensive ground-based study has been carried out there and the results are in preparation for publication (Jagoda et al., in preparation, 2024).



For comparison, we repeated the analysis using the EDGAR inventory to estimate non-
O&G sources for the large regions (see SI, Table S6). After removing double counting and
adjusting for emissions from other sources as described previously, the total emissions
measured attributed to O&G production are $12,732 \pm 4932$ kg hr$^{-1}$. This is slightly lower than
the total emissions estimated using the TNO-CAMS inventory $13,216 \pm 4932$ kg hr$^{-1}$, indicating
a larger fraction of non-O&G sources in the EDGAR inventory. The inferred O&G emissions,
taking into account the non-O&G emissions from the EDGAR inventory result in a facility-
weighted emission factor of $5.1 \pm 2.0$ kg hr$^{-1}$ site$^{-1}$, consistent with the $5.3 \pm 2.0$ kg hr$^{-1}$ site$^{-1}$
when using TNO-CAMS for the non-O&G sectors. It is important to note that the inventory
estimates for the non-O&G sectors do not differ strongly between EDGAR and TNO-CAMS
in the regions where we apply the corrections. However, this is not the case for all regions in
the southern Romanian production basin. Table S7 in the Supplement shows that the
discrepancies between the two inventories can become large. Specifically, in EDGAR, the non-
O&G emissions are higher than those in TNO-CAMS, nearly double in some cases. Moreover,
O&G emissions are very low in EDGAR, whereas they contribute to almost half of the
emissions in TNO-CAMS. Because of this more balanced contribution from all sources, we use
the estimates from TNO-CAMS for our central emission factor estimate and for the upscaling.
*Table 1 - Measured emission rates (ER) and estimates of the O&G related fraction of total CH$_4$*
*emissions in target regions and clusters. "Non-O&G emissions (kg hr$^{-1}$)" are extracted from*
*the TNO-CAMS inventory for the target regions and are used to derive ERs from the O&G*
*industry in the area ("O&G emissions"). The last column shows the emission factor (kg CH$_4$*
*hr$^{-1}$ site$^{-1}$). Numbers in bold are used for upscaling to the national scale (see text for details).*

| Flight ID | Target region/cluster | # facilities | # wells | Total Measured Emissions (kg hr$^{-1}$) | Non-O&G emissions (kg hr$^{-1}$) | O&G emissions (kg hr$^{-1}$) | EF (kg h$^{-1}$ site$^{-1}$) |
|---|---|---|---|---|---|---|---|
| SA01 | R7 | 496 | 337 | $8517 \pm 2097$ | 1388 | $7129 \pm 2097$ | $14.4 \pm 4.2$ |
| SA02 | R7 | 504 | 343 | $8335 \pm 1440$ | 1388 | $6947 \pm 1440$ | $13.8 \pm 2.9$ |
| SA03 | R5a | 827 | 654 | $4556 \pm 2570$ | 772 | $3784 \pm 2570$ | $4.6 \pm 3.1$ |
| SA04 | R5a-small | 818 | 642 | $2920 \pm 935$ | 374 | $2516 \pm 935$ | $3.1 \pm 1.1$ |
| SA05 | R6C2C3C4 | 471 | 379 | $1729 \pm 912$ | - | $1729 \pm 912$ | $3.7 \pm 1.9$ |
| SA06 | R7C3C4 | 124 | 92 | $1481 \pm 287$ | - | $1481 \pm 287$ | $11.9 \pm 2.3$ |
| SA07 | R7C2 | 71 | 44 | $1395 \pm 546$ | - | $1395 \pm 546$ | $19.6 \pm 7.7$ |
| SA08 | R7VentArea | 67 | 41 | $602 \pm 209$ | - | $602 \pm 209$ | $9.0 \pm 3.1$ |
| SA09 | R4C5 | 390 | 347 | $477 \pm 106$ | - | $477 \pm 106$ | $1.2 \pm 0.3$ |
| SA10 | R6C6 | 29 | 16 | $469 \pm 170$ | 130† | $339 \pm 170$ | $11.7 \pm 5.9$ |
| SA11 | R7Vent | 37 | 20 | $266 \pm 113$ | - | $266 \pm 113$ | $7.2 \pm 3.1$ |
| SA12 | R7C5 | 59 | 45 | $259 \pm 47$ | - | $259 \pm 47$ | $4.4 \pm 0.8$ |
| SA13 | R4C2C3 | 247 | 186 | $246 \pm 89$ | - | $246 \pm 89$ | $1.0 \pm 0.4$ |
| SA14 | R6C5 | 27 | 21 | $131 \pm 85$ | - | $131 \pm 85$ | $4.9 \pm 3.1$ |
| SA16 | R8C1 | 29 | 19 | $90 \pm 49$ | - | $90 \pm 49$ | $3.1 \pm 1.7$ |
| SA17 | R7C8 | 48 | 43 | $78 \pm 101$ | - | $78 \pm 101$ | $1.6 \pm 2.1$ |
| SA18 | R7C1Facility | 8 | 5 | $13 \pm 9$ | - | $13 \pm 9$ | $1.6 \pm 1.1$ |
| Weighted mean, everything | | 4358 | 3303 | $31667 \pm 10039$ | *, † | $27513 \pm 9765$ | $6.3 \pm 2.2$ |



| No double counting | 2516 | 1956 | | *, †, †† | **13216 ± 4932** | **5.3 ± 2.0** |
|---|---|---|---|---|---|---|
| Sum of clusters in R7 | 377 | 270 | | | 3828 ± 1199 | 10.2 ± 3.2 |
| Only clusters with 100% fossil | 1570 | 1238 | | | 6970 ± 2610 | 4.4 ± 1.7 |

\* considering the absolute non-O&G emissions from the TNO-CAMS inventory for the large regions and 100% O&G contribution for the clusters

† accounting for landfill within R6C6

†† excluding cluster quantifications in R7

### 3.2. Qualitative information from measurement - simulation comparisons
#### 3.2.1. Example comparison of meteorology and CH₄ for a mass balance flight

Figure 4 shows an example of a comparison between measurements along the SA mass balance flight from October 17, 2019, with results from the COSMO-GHG and MECO(3) models. The top two panels show simulated and measured $CH_4$ mole fractions along the flight track and the bottom two panel shows the vertical $CH_4$ profiles in the simulations along the flight track above the changing orography (black). During this flight, 4 different clusters and combinations of clusters were circled multiple times at different altitude; the flight altitude is included in the bottom panels as dashed black line. The repeating orographic patterns guide the eye in following the circular flight patterns around the clusters and are numbered in white. The colored contours illustrate the vertical $CH_4$ profiles along the flight track. The measured plume in the first, largest, cluster is captured relatively well by the simulation for some of the cycles, but during some cycles the flight track is partly above the boundary layer in the models and the peak is not fully captured. During cycles 4 and 5, the observations suggest that the aircraft was flying above the boundary layer also in reality, but one sharp, narrow peak was still observed after the highest orographic peak in the measurements, which is missing in the simulation. For the second cluster that was cycled 12 times, the COSMO-GHG model captures the plumes better than the MECO(3) model. For both models, the simulated and measured $CH_4$ mole fractions show a consistent transition out of the boundary layer in cycles 7-9, indicating a good representation of the boundary layer height in the models. For the third cluster, the models are missing the large, sharp peaks, indicating missing emissions in this cluster. In addition, the MECO(3) model simulates higher plumes when the flight track was in the model boundary layer, but lower plumes when the flight track was outside the boundary layer. For the last cluster, the simulated and measured elevations are small and relatively consistent for COSMO-GHG, but the MECO(3) model simulates some larger plumes spanning more than one cycle, indicating larger scale upwind contamination, which was also documented in the observations.

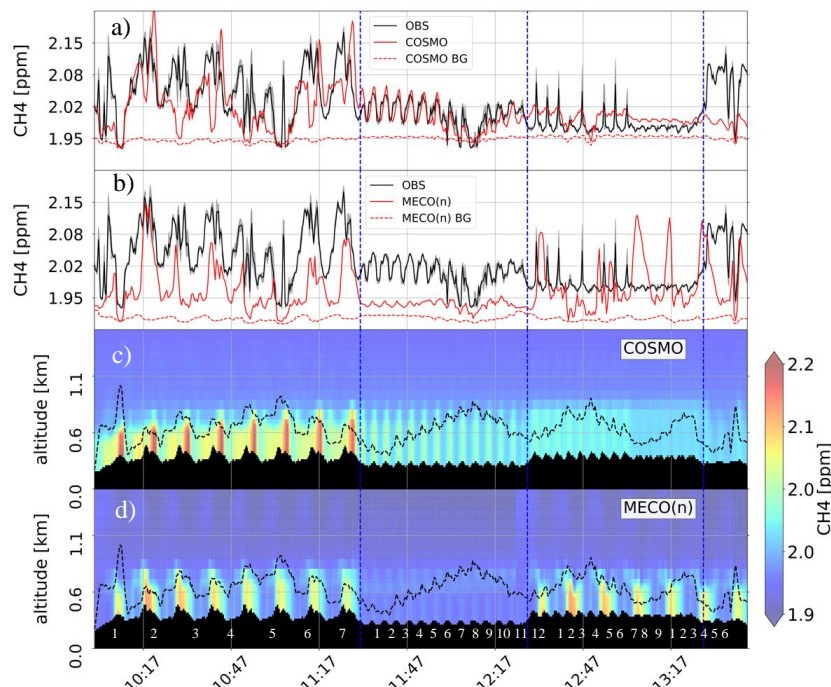

*Figure 3 – Measurements and simulation results of (a&b) CH₄ mole fraction along the flight*
*track, and (c&d) the vertical CH₄ profile along the flight track as simulated by the COSMO*
*model (a&c) and the MECO(3) model. Model background fields are shown as dashed lines in*
*a&b. Panel c&d also include the flight track as black dashed line, and the black contour at the*
*bottom shows the orography in this mountainous terrain; the repeating patterns illustrate*
*individual cycles around the clusters R6C2C3C4, R6C5, R6C6 and R6C7, cycles are numbered*
*in white. The flight around cluster R6C7 did not allow successful emission quantification*
*because of an upwind influence and is therefore not included in Table 1.*
A similar analysis was performed for each flight with the goal to identify plumes where either
the simulation results or the measurements indicated that the respective circle was flown outside
the simulated or actual boundary layer. In this case, the respective plume was not retained for
the measurement – simulation comparison. In total, 10 out of 200 individual plumes were
rejected this way. In addition, 66 circles around clusters that were influenced by signals from
upwind sources were excluded.
**3.2.2. Model performance in terms of meteorology**
As mentioned above, the low winds during the campaign period presented difficult
meteorological conditions for emissions quantification. We performed a thorough
meteorological analysis to identify days when the meteorological conditions agree well between
the two models and the measurements. The results are shown in S.1 in the SI, which illustrate
that it was not possible to identify days when the meteorological conditions agree well between
the two models and the measurements. Therefore, it was decided not to focus on individual days
or flights. Rather, in the following we compare the measured and simulated plume areas
statistically across all available flights. This is done to investigate whether correlation of
measured and simulated CH₄ enhancements from the raster flights, which cover a wider region,



is similar to the one for the individual plumes quantified during the mass balance flights. The
analysis, which is described in the section below, can also possibly identify regional differences
and be used to derive approximate scaling factors for the raster flights in comparison with the
mass balance flights.

### 3.3. Measurement - model comparison of plume areas for mass balance flights

We first evaluate individual plume-level data from the mass balance flights, because for
these flights we have measured emission rates from the mass balance approach. Thus, we can
compare the measured and simulated plume areas and derive a correction factor for the emission
rates assumed in the model that would bring the measured and simulated plumes to agreement.
Fig. 4 shows the comparison of the observed and simulated plume areas for 190 individual
plumes evaluated from the SA mass balance flights and COSMO-GHG and MECO(3) models,
respectively.
For mass balance flights around production clusters, each circle around a cluster results
in one or few down-wind plumes (which are integrated in our analysis), but for mass balance
flights targeting larger regions, numerous well-separated plumes can generally be quantified
from a single circle. The high scatter in the comparison between simulated and measured plume
areas can be ascribed to a number of factors, for example: i) large variability in actual emissions
from different source areas (here: production clusters), including the important role of super
emitters, ii) difficult meteorological conditions with low wind leading to variable plume
representations, both in the real atmosphere and in the model, iii) over- or underestimates
associated with the dynamics of the planetary boundary layer, and iv) variable measurement
distance from the emission points. The scatter in the comparison of plume areas with MECO(3)
results is even larger than for the COSMO-GHG model. This is ascribed to the fact that the
meteorological fields in COSMO-GHG are nudged to observations, whereas MECO(3) nudges
only the global model instance, implying more degrees of freedom within the nested instances
to develop their own (sub-synoptic) meteorological situation which might deviate from the data
used for nudging. Indeed, the meteorological evaluation (See S.1 in the SI) shows that the
meteorological fields in COSMO-GHG (directly nudged) are closer to the observed
meteorological parameters than for MECO(3), as expected.

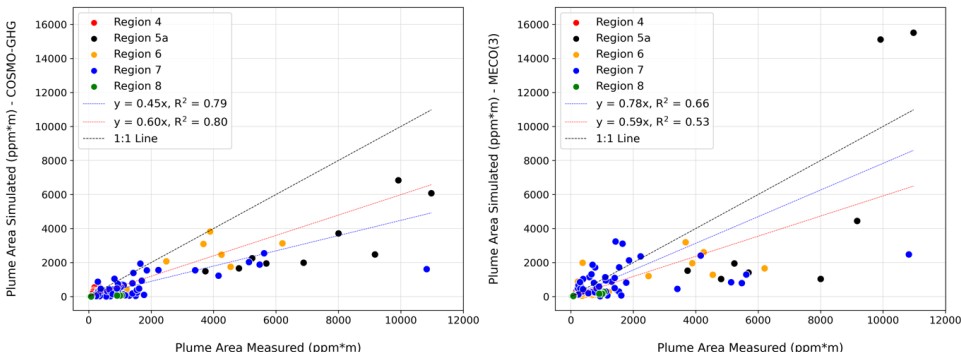

*Figure 4 - Comparisons between plume areas calculated from measurements and simulations with COSMO-GHG (left) and MECO(3) (right). Blue dashed lines show linear fits to all data and red dashed lines linear fits to the plumes from the clusters only, without the points from the larger regions. Plots zooming in on the region of plume areas up to 2000 ppm * m are shown Figure S1 in the SI.*



Nevertheless, despite the variability, it is evident that most of the points fall well below
the 1:1 line, which means that the simulated plume areas along the flight track that were
generated with an assumed emission factor of 1 g s$^{-1}$ site$^{-1}$, thus 3.6 kg hr$^{-1}$ site$^{-1}$, generally
underestimate the measured plume areas. The further the points fall below the 1:1 line, the
higher the implied mismatch in the emission rate that was assumed in the model. A linear fit to
all the measured and simulated plumes has a slope of 0.44 for COSMO-GHG, and 0.78 for
MECO(3). When we exclude the points from the larger regions, where the measured plumes
are often further away from the source regions, the slopes change slightly to 0.56 for COSMO-
GHG, and to 0.62 for MECO(3). This suggests that the assumed emission rate in the model is
on average underestimated by about a factor of 2. However, quantitative interpretation is
problematic in this approach, since the slope of the linear fit is largely determined by a relatively
small number of plumes with large plume areas. Furthermore, the sampling is biased towards
clusters where more circles were flown (i.e., circles at more altitude levels), and does not
consider the number of facilities per cluster. In addition, there may be systematic biases in the
models, e.g. due to model resolution or meteorological conditions (as discussed above), that
lead to smaller plume areas in the models compared to the measurements. For the present
purpose, we will compare the slope of observed and simulated plume areas from the mass
balance flights determined here with the slope of observed and simulated $CH_4$ enhancements
from the raster flights in section 3.4.2 to investigate whether the enhancements observed during
the raster flights qualitatively agree with the ones from the mass balance flights.

### 3.4. Measurement - model comparison of plume areas for raster flights

Figure 5 shows the comparison of the integrated enhancement above background along
the flight tracks for $CH_4$ mole fractions measured during the raster flights and simulated with
the two models. The scatter for these integrated enhancements is smaller than for the individual
plume areas shown in Fig 4., which likely reflects the fact that the integrated enhancements are
the sum of numerous plumes, and high and low values average out for the integrated
enhancements.

Similar to the plume area comparison from the mass balance flights (Fig. 4), most of the
points fall below the 1: 1 line, again indicating that the emission rate of 3.6 kg hr$^{-1}$ site$^{-1}$ assumed
in the models is insufficient to explain the observed concentrations. The slopes of the
orthogonal linear regressions of 0.43 and 0.33 for the two different models are even lower than
for the mass balance flights above, indicating a possible underestimate by up to a factor of 3 in
the assumed emission rate. Still, the slopes are in a similar range as the slopes from the mass
balance flights in Fig. 4. It is important to note that these slopes were now derived from the
simulated fields under similar conditions as the ones for the individual plumes from the clusters.
Thus, whereas various factors could cause systematic under- or overestimates in simulated
versus measured $CH_4$ enhancements, the similar slopes obtained for the two types of flights
suggest that the emission characteristics of the plumes observed during the mass balance and
raster flights are compatible. Thus, the emission factors derived for a limited number of clusters
in section 3.3 are likely representative for the larger areas covered in the mass balance flights,
and thus for a large fraction of the Southern Romanian O&G production infrastructure. We
conclude that the $CH_4$ enhancements observed on the BN2 aircraft during the raster flights
generally support the emission factors derived in section 3.1 from the mass balance approach.



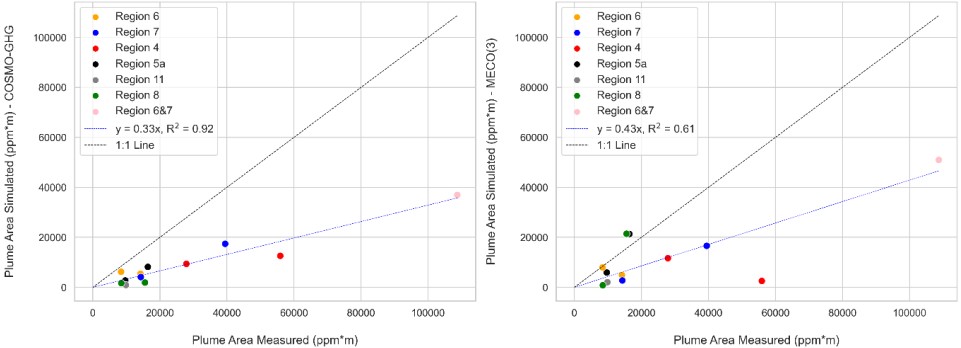

*Figure 5 - Comparison between integrated CH₄ enhancements from measurements during raster flights on the BN2 aircraft, and simulations along the flight tracks with COSMOS-GHG (left) and MECO(3) (right). Different colors represent different regions. Linear fits to the data are shown as blue dashed lines and the 1: 1 line is shown as black dashed line.*

## 4. Conclusions

Airborne measurements of methane performed from two aircraft during the ROMEO 2019 campaign were evaluated to obtain emission rate estimates representative for production clusters and larger regions in the O&G production basin in Southern Romania. Emissions determined from a mass balance approach yield a wide range of instantaneous emission factor estimates between different clusters, supporting the heterogeneity of emissions across individual sites, regions and time. Assessment of the O&G emissions from flights around larger regions is difficult because of the unknown contribution of emissions from other sectors. From mass balance estimates covering a total of 2516 sites, using the TNO-CAMS inventory to derive emissions from non-O&G sources for the large regions, and assuming 100% of the observed emissions in the smaller clusters to originate from O&G production, we derive total emissions of $13{,}216 \pm 4932$ kg hr$^{-1}$ for the covered regions in Southern Romania. This results in a facility-weighted emission factor of $5.3 \pm 2.0$ kg hr$^{-1}$ site$^{-1}$, consistent with the previously published estimate from ground-based quantifications of 5.4 kg hr$^{-1}$ oil production site$^{-1}$ (range $3.6 - 8.4$ kg hr$^{-1}$ site$^{-1}$, (Stavropoulou et al., 2023). The facility-weighted average for 1570 facilities in dense production clusters, where we are certain that the dominant contribution is from the O&G infrastructure is $4.4 \pm 1.7$ kg hr$^{-1}$ site$^{-1}$, aligning with the estimate from larger regions. Using the of EF 5.3 kg hr$^{-1}$ site$^{-1}$ to scale up to the national scale results in an annual emission rate estimate of $344 \pm 130$ ktons CH₄ yr$^{-1}$, which is about three times higher than the UNFCCC reported national emissions from the O&G industry for Romania. Mole fraction measurements carried out in raster flight tracks over wider areas lacked meteorological measurements and therefore could not be used to derive direct estimates of emission rates. To support the evaluation, simulations with two numerical atmospheric models were carried out and the simulated CH₄ fields were compared with the measurements. Due to the difficult meteorological conditions, direct quantitative evaluation remains challenging, but the comparison of observed and simulated enhancements consistently suggests that the prior emission rate of 3.6 kg hr$^{-1}$ site$^{-1}$ used in the models is too low. In addition, the correlation of measured and simulated CH₄ enhancements for the raster flights over larger areas is consistent with the correlations observed in mass balance flights around well-defined production clusters, indicating the validity of the derived emission factors for a large part of the southern Romanian O&G production region. We conclude that the top-down emission estimates derived here from airborne surveys over larger regions support the previously published emission rate estimates



derived from ground-based bottom-up quantifications during the ROMEO 2019 campaign.
These results confirm that O&G methane emissions in 2019 were much higher than reported to
UNFCCC.
**Data availability.** In-situ measurements and outputs of model simulations along flight tracks
are available from Maazallahi et al. (2024a).
**Code availability.** MATLAB® codes for investigation of in-situ measurements from circular-
pattern and raster flights and outputs of model simulations are available from Maazallahi et al.
(2024b).
**Acknowledgements.** The ROMEO project was supported by the Climate and Clean Air
Coalition (CCAC) Oil and Gas Methane Science Studies (MMS) hosted by the United Nations
Environment Programme UNEP. Funding was provided by the Environmental Defense Fund,
Oil and Gas Climate Initiative, European Commission, and CCAC. This project received
further support from the H2020 Marie Skłodowska-Curie project Methane goes Mobile –
Measurements and Modelling (MEMO[2]; https://h2020-memo2.eu/), grant number 722479. The
modeling work used resources of the Deutsches Klimarechenzentrum (DKRZ) granted by its
Scientific Steering Committee (WLA) under project ID bd0617 to perform the MECO(3)
simulations.
**Competing interests.** At least one of the (co-) authors is a member of the editorial board of
Atmospheric Chemistry and Physics. The authors have no other competing interests to declare.
**Author contributions.** MLS, SAC, SG, AP, and MA, carried out and evaluated airborne
measurements, HM carried out the quantitative data evaluation, SJS carried out the
meteorological analysis, FS supported the data analysis, MS, DB, MM, PJ, carried out the
atmospheric simulations, AV, HDvdG, SD, and NVS provided inventory information, SS, MA,
AC and TR designed and planned the study, HM and TR drafted the manuscript.

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

Methods to Reduce Emissions, on the Path to the Paris Agreement, Reviews of Geophysics,
58, 2020.
Röckmann, T., Eyer, S., van der Veen, C., Popa, M. E., Tuzson, B., et al.: In situ observations of
the isotopic composition of methane at the Cabauw tall tower site, Atmos. Chem. Phys.,
16, 10469-10487, 10.5194/acp-16-10469-2016, 2016.
Saarnio, S., Winiwarter, W., and Leitão, J.: Methane release from wetlands and watercourses
in    Europe,    Atmos.    Environ.,    43,    1421-1429,
https://doi.org/10.1016/j.atmosenv.2008.04.007, 2009.
Saunois, M., Stavert, A. R., Poulter, B., Bousquet, P., Canadell, J. G., et al.: The Global Methane
Budget 2000-2017, Earth Syst Sci Data, 12, 1561-1623, 10.5194/essd-12-1561-2020, 2020.
Shindell, D., Ravishankara, A. R., Kuylenstierna, J. C. I., Michalopoulou, E., Höglund-Isaksson,
L., et al.: Global MethaneAssessment: Benefits and Costs of Mitigating Methane Emissions,
United Nations Environment Programme and Climate and Clean Air Coalition, Nairobi:
United Nations Environment Programme., 2021.
Stavropoulou, F., Vinković, K., Kers, B., de Vries, M., van Heuven, S., et al.: High potential for
CH₄ emission mitigation from oil infrastructure in one of EU's major production regions,
Atmos. Chem. Phys., 23, 10399-10412, 10.5194/acp-23-10399-2023, 2023.
Szopa, S., Naik, V., Adhikary, B., Artaxo, P., Berntsen, T., et al.: Short-lived Climate Forcers, in:
Climate Change 2021 – The Physical Science Basis: Working Group I Contribution to the
Sixth Assessment Report of the Intergovernmental Panel on Climate Change, edited by:
Intergovernmental Panel on Climate, C., Cambridge University Press, Cambridge, 817-922,
2021.
UNFCCC: Paris Agreement to the United Nations Framework Convention on Climate Change,
T.I.A.S. No. 16-1104, 2015.
UNFCCC:    Greenhouse    Gas    Inventory    Data—Comparison    by    Gas.
https://di.unfccc.int/comparison_by_gas, 2023., 2023a.
UNFCCC,    Romania.    2023    Common    Reporting    Format    (CRF)    Table,
https://unfccc.int/documents/627660 2023b



Weller, Z. D., Hamburg, S. P., and von Fischer, J. C.: A National Estimate of Methane Leakage
from Pipeline Mains in Natural Gas Local Distribution Systems, Environmental Science &
Technology, 54, 8958-8967, 10.1021/acs.est.0c00437, 2020.
Winterstein, F., and Jöckel, P.: Methane chemistry in a nutshell – the new submodels CH4
(v1.0) and TRSYNC (v1.0) in MESSy (v2.54.0), Geosci. Model Dev., 14, 661-674,
10.5194/gmd-14-661-2021, 2021.
