# Peer review of "Airborne in-situ quantification of methane emissions from oil and gas production in Romania"

_EGUsphere, 2024_

## Author Comment (AC1)

We are very much thankful for the constructive comments received from the anonymous referee, those were useful in improving the manuscript. Please find our replies in blue and changes applied in the revised manuscript in ***bold italic blue*** letter style.

‒ ‒ ‒ ‒ ‒ ‒ ‒ ‒ ‒ ‒ ‒ ‒ ‒ ‒ ‒ ‒ ‒ ‒ ‒ ‒ ‒ ‒ ‒ ‒ ‒ ‒ ‒ ‒ ‒ ‒ ‒ ‒

Referee Comments 1 (RC1) for the manuscript titled:

**Airborne in-situ quantification of methane emissions from oil and gas production in Romania**

Maazallahi, H., Stavropoulou, F., Sutanto, S. J., Steiner, M., Brunner, D., Mertens, M., Jöckel, P., Visschedijk, A., Denier van der Gon, H., Dellaert, S., Velandia Salinas, N., Schwietzke, S., Zavala-Araiza, D., Ghemulet, S., Pana, A., Ardelean, M., Corbu, M., Calcan, A., Conley, S. A., Smith, M. L., and Röckmann, T.

***Correspondence to***:

Hossein Maazallahi (h.maazallahi@ut.ac.ir), Thomas Röckmann (t.roeckmann@uu.nl)

*This manuscript is one of the outputs of the ROMEO campaign.*

‒ ‒ ‒ ‒ ‒ ‒ ‒ ‒ ‒ ‒ ‒ ‒ ‒ ‒ ‒ ‒ ‒ ‒ ‒ ‒ ‒ ‒ ‒ ‒ ‒ ‒ ‒ ‒ ‒ ‒ ‒ ‒

Airborne in-situ quantification of methane emissions from oil and gas production in Romania presents results from the airborne part of the 2019 ROMEO campaign in Romania. It takes advantages of the numerous flights around more or less large areas to infer methane emissions and emissions factors for these regions and extrapolated to the country. The authors detail the assumptions and limitations of their work clearly and make use of all the data to make them as robust as possible. I recommend publication after minor corrections.

**Comments:**

Figure1: please add the name of the clusters on the map, also maybe in the SI a table with the number of flights for each cluster and regions, dates, would help understand the results later on.

We have now added the cluster numbers within each respective larger region in Figure 1 and flight dates are added in Table S6.

Section 2.2

Can you elaborate on the quality procedure for the measurements? Though calibration is not necessary for the biases as you deal with differences to background, were there any check for time-drift maybe especially with the AERIS instrument which may not be as stable as PICARRO instrument usually are.

The delay times for the instruments were checked and corrected. Instruments were calibrated before the campaign, but not regularly during the campaign. Indeed, the Picarro analyzer is very stable. For the quantification analyses, in-situ measurements from the stable

Picarro analyzer were used, which is now added (in bold italic) in the manuscript as follows (See L271):

> For the mass balance flights (Fig 2a), the lowest $CH_4$ value of each circle around a target area ***retrieved from the Picarro instrument*** was defined as background mole fraction and subtracted from downwind measurements to obtain the $CH_4$ enhancement.

As no met data were measured on the raster flight, how did you check the prevailing wind direction and checked that it didn't change during the flight time?

> The wind directions were derived from the model output analyzed for this paper, please see Supplementary S1. As stated in our paper, the uncertainty in wind speed is one of the key uncertainties for the quantitative evaluation. Wind direction has less impact on the measurements for the raster flights relative to the circular flights as the latter were used for the quantitative analysis. The raster flights were close to the ground, but the distance to the sources is not always clear. This leads to a large uncertainty for quantitative evaluation, which is why we evaluated the results only in a statistical manner, not peak-to-peak.

Section 2.3 l 180-186 this passage is not clear and these tracers are not talked about afterwards. Need to clarify.

> The initially used word tracers in Sect. 2.3 refers to the representative points of clusters and regions were 1 kg hr$^{-1}$ methane was released in the simulations. In the manuscript we focused on the comparison between the measurements and outputs of the simulations in which model-based prognostic $CH_4$ tracers analyses were not directly included. Here we provide the information to describe the simulations settings. As this term can bring confusion with the commonly use of tracer for emission attribution purpose, we have changed it to explanatory explanation of model-based prognostic $CH_4$ tracer The text in Sect. 2.3 is now edited as follows (See L184-L192): ***To be able to geo-attribute emissions to certain emission clusters, we applied 33 individual model-based prognostic $CH_4$ tracers in the models which are transported according to the meteorological conditions. Each of these tracers represents the emissions of a specific area with a fixed emission rate of 1 g s$^{-1}$ or 3.6 kg hr$^{-1}$ and released at one individual or multiple release point(s). Meaning that one tracer represents the emissions of one or two clusters and one or two distant regions, assuming that they are sufficiently far away. This allows us to separate the signal of each cluster / region flown over or circled around. During the analysis, these tracers are not further considered, because, since the attribution by location is usually unambiguous.***

Section 2.7.: can you add an equation or a figure to illustrate the integration of the measurement along the flight path and the way of calculating the emissions from them?

The following text is now added to the manuscript in Sect. 2.7.1 (See L262-L268):

> ***Equation 1 is used to translate the aircraft measurements into the emission rates which is described in details by Conley et al. (2017).***

$$Q_c = \left(\frac{\partial m}{\partial t}\right) + \int_0^{zmax} \oint c' u_h \cdot \hat{n} \, dl dz \qquad\qquad Eq.\ 1$$

***Here, $Q_c$ is the net emission from source (s) and sink (s), l is the position along the flight path, $\hat{n}$ is the a vector normal to surface pointing outward, $u_h$ (= $u_i$ + $v_j$ ), c' is the $CH_4$ enhancement from the mean of each circle's mixing ratio and $\left(\frac{\partial m}{\partial t}\right)$ is the total mass trend within the volume of each box.***

How did you identify the up-stream contaminations?

During the circular flights, the up-stream contaminations were excluded from the measurements-based quantifications using wind direction. Manually speaking, this exclusion was based on the wind direction, meaning that if the wind direction was toward the inside of a box and $CH_4$ enhancements were observed at the time during flights, those enhancements were flagged as contamination from outside, hence excluded. However, this process is automatically done by defining the vector $\hat{n}$ in the Eq. 1.

Section 3.1l314-343: this is not clear, what is weighted and how, why don't you just average for the sites you measured several times? the passage has to be rephrased, streamlined and maybe have a basic equation to show what are you weighting and how.

When a site, e.g. site A, is measured several times and others are measured once, the average of all measurements is biased toward the site A. Here we have similar situation. After removing the double and triple counts and non-O&G sources, the EF from the regions reaches $5.3 \pm 2.0$ kg $hr^{-1}$ $site^{-1}$. That's right, we apologize for the wording and unnecessary explanation of weights for the regions / cluster. This is now removed because the EFs are derived based on normal averaging. The lines were rephrased as follows (See L327-L370):

*The sum of all emissions from the airborne$CH_4$ emission measurements (SA01-SA18) from all flights reach 31,700 kg $hr^{-1}$ accounting for 4358 active sites measured during all flights combined (Table 1). This results in EF of 7.3 kg $hr^{-1}$ $site^{-1}$ after a simple division. However, this EF is biased for two reasons: (I) not all emissions measured (31,700 kg $hr^{-1}$) are from O&G sources and (II) there are double to triple countings of emissions in total sum, e.g. R5a and R7 is measured twice or three times. The first point results in overestimation of EFs from O&G activities and the latter point results in biasing the average EF towards emission rates of sources which were measured more than once. Therefore, we performed several analyses to address these two points.*

*In total, in addition to cluster-focused flights for R7, two regional flights have been performed per R7 and R5a each, which results in triple countings of emissions for R7 and double countings of emissions for R5ain the total sum of 31,700 kg $hr^{-1}$. Hence, we used average emission rates from the regional measurements targeting the R7 and R5a individually (SA01 and SA02 for R7 and SA03 and SA04 for R5a, respectively). For the regions R4, R6 and R8 no regional flights were performed, and cluster-focused quantifications were performed. We used the sum of emissions from these clusters as the total emissions for these regions. These corrections result in cumulative emissions of 13,200 ± 4,932 kg $hr^{-1}$ for these regions, accounting for 2516 active sites which results in EF of 5.3 ± 2.0 kg $hr^{-1}$ $site^{-1}$.*

*Acting on the field observations and inventory information, emissions from all clusters can be assigned to O&G activities except for the R6C6. After deducting reported emissions for the landfill within the boundary of R6C6 and adding to the measured emissions from other clusters, we reach total emission of 6,970 ± 2,610 kg $hr^{-1}$ for 1,570 sites which results in EF of is 4.4 ± 1.7 kg $hr^{-1}$ $site^{-1}$.*

*Both EFs, 5.3 ± 2.0 kg $hr^{-1}$ $site^{-1}$ and 4.4 ± 1.7 kg $hr^{-1}$ $site^{-1}$, overlap with the EF of 5.4 kg $hr^{-1}$ (95% CI: 3.6 – 8.4 kg $hr^{-1}$) oil production $site^{-1}$ reported from ground-based measurements by Stavropoulou et al. (2023). However, both EFs from the airborne measurements fall on the lower side of the EF from the ground based measurement. This*

*could be explained as follows: (I) It is assumed in Eq. 1 that all emissions within the flight boundaries are transported horizontally and captured during the flights. However, during the ROMEO campaign, the low wind speed condition and high solar radiation could result in vertical transport, which was not measured during the airborne measurements. It is possible that the area mass balance quantifications in the flat and arid region R5a in Southern Romania may be biased slightly low due to partial loss of $CH_4$ out of the boundary layer during the hot and convective conditions, or due to the fact that stable transport conditions had not yet established over the large regions. (II) The quantifications reported by Stavropoulou et al.(2023) were focused on the oil production for which gas production, which is mostly methane, is not favorable, hence released which we could also observe through optical gas imaging cameras. This release is favorable to happen at the production sites to prevent two-phase conditions in the pipelines and collection and processing systems. These two reasons individually or combined could explain this average difference between the EFs derived from airborne and ground-based measurements. The difference between the two EFs derived from the airborne measurements, $5.3 \pm 2.0$ kg hr$^{-1}$ site$^{-1}$ from regional measurements and $4.4 \pm 1.7$ kg hr$^{-1}$ site$^{-1}$ from the clusters only, could be explained by the presence of large emitters outside the clusters but within the regional boundaries.*

**Minor comments:**

l57: substancial instead of substation

Done.

l60: emission meaqurements

Corrected.

l88: what of the 2nd phase?

The following sentence is now added to the manuscript (See L92-L95):

*The second phase happened in the following year and focused on the gas production region in the Transylvanian Basin, north of the mountain range.*

l104: production asset

Corrected.

l106 remove 'total', replace 'where' by 'though'

Done.

l119: remove the last sentence or add the black symbol

The sentence is now removed.

l144: as above

The sentence is now removed.

l 168: remove () around the citation

Removed.

l173: add space after 2021)

Added.

l282: emission quantifications

Corrected.

l 288 remove 'to'

Removed.

l352: remove 'about'

Removed.

l357 remove () around citation

Paid attention to during the rephrasing of the paragraph.

l370: replace 'slights' by 'flights'?

Replaced.

l381: replace 'estimated at' by 'reached'

Done.

l415: replace Figure 4 by Figure 3

Done.

l565: 'EF of'

Corrected.

**Reference**

Stavropoulou, F., Vinković, K., Kers, B., de Vries, M., van Heuven, S., Korbeń, P., Schmidt, M., Wietzel, J., Jagoda, P., Necki, J. M., Bartyzel, J., Maazallahi, H., Menoud, M., van der Veen, C., Walter, S., Tuzson, B., Ravelid, J., Morales, R. P., Emmenegger, L., Brunner, D., Steiner, M., Hensen, A., Velzeboer, I., van den Bulk, P., Denier van der Gon, H., Delre, A., Edjabou, M. E., Scheutz, C., Corbu, M., Iancu, S., Moaca, D., Scarlat, A., Tudor, A., Vizireanu, I., Calcan, A., Ardelean, M., Ghemulet, S., Pana, A., Constantinescu, A., Cusa, L., Nica, A., Baciu, C., Pop, C., Radovici, A., Mereuta, A., Stefanie, H., Dandocsi, A., Hermans, B., Schwietzke, S., Zavala-Araiza, D., Chen, H., and Röckmann, T.: High potential for CH4 emission mitigation from oil infrastructure in one of EU's major production regions, Atmos. Chem. Phys., 23, 10399–10412, https://doi.org/10.5194/acp-23-10399-2023, 2023.

---

## Author Comment (AC2)

We highly appreciate the constructive comments of the anonymous referee, those were useful to improve the manuscript. Please find our comments in blue and changes applied in the revised version of the manuscript in ***bold italic blue*** letter style.
* * *
Referee Comments 1 (RC2) for the manuscript titled:

**Airborne in-situ quantification of methane emissions from oil and gas production in Romania**

Maazallahi, H., Stavropoulou, F., Sutanto, S. J., Steiner, M., Brunner, D., Mertens, M., Jöckel, P., Visschedijk, A., Denier van der Gon, H., Dellaert, S., Velandia Salinas, N., Schwietzke, S., Zavala-Araiza, D., Ghemulet, S., Pana, A., Ardelean, M., Corbu, M., Calcan, A., Conley, S. A., Smith, M. L., and Röckmann, T.

***Correspondence to***:

Hossein Maazallahi (h.maazallahi@ut.ac.ir), Thomas Röckmann (t.roeckmann@uu.nl)

*This manuscript is one of the outputs of the ROMEO campaign.*
* * *
This is a very interesting manuscript that forms one of the outputs from the ROMEO project surveys of 2019. The low-wind conditions at the time were very challenging for flight surveys, but the authors have managed to tease out some important conclusions. The overall finding is that there is good agreement between aircraft and ground surveys, so it would be useful to have a concluding statement about why aircraft should be used in ground-accessible locations for this type of survey going forward, given the relative cost implications and the meteorological challenges. The comparison with the ground surveys already published was quite cursory and could be developed further, as all these surveys should be producing an emission per facility and this is the main output for comparison with the model and inventories.

The following text has been added to the conclusion following the recommendation regarding the use of airborne measurements for the areas where ground surveys can also be conducted (See L599-L605):

*Airborne measurements for the regions and clusters, where ground-based surveys can be also applied, can provide important additional insight, such as: (I) the influence of super emitters is included as a realistic fraction in the total airborne measured emissions while super emitters may be either missed or accidentally be overrepresented in ground surveys, (II) the influence of non-O&G sources on total emission can be studied, and (III) airborne quantification can cover large areas in a much shorter time compared to ground-based quantification.*

**Detailed Comments:**

Line 51 – There is a new version of the Saunois et al paper for 2024.

Indeed, however the paper is still in the discussion phase, so we refer to the fully peer-reviewed paper for now.

Line 57 – substation? I think that you mean 'substantial'

This is now corrected.

Line 78 – I was not aware of such a significant O&G sector in Poland, so I am just double-checking that this is not a high annual emission for all fossil fuels including coal.

Poland is indeed primarily known for its coal-related emissions, as it ranks first in coal production within the EU. However, according to IEA data, Poland also ranks 4th in natural gas production and 6th in oil production among EU countries. These factors contribute to Poland having the second-highest annual emissions in the EU for the category 1.B.2 Oil and Natural Gas and Other Emissions from Energy Production as of 2021.

Figure 2 – Can you add wind direction to the figures or include in the captions. It is not easy to locate 2a on figure 1 without the region 7 bounding box. Also there is no explanation of why the lowest concentrations measured during the regional surveys are 2.67 ppm? Is this related to the low wind speeds mentioned during regional surveys, or instrument calibration. If the latter it would be better to display the data as an excess over baseline.

We recognize the challenge in identifying the locations of the clusters within each larger region. In response to your comment and the first referee's request, we have now included in Figure 1 the cluster numbers within each respective larger region for clarity. Explanation of wind directions was added in the caption of Figure 2.

The instrument was calibrated for the flights. From the raster flight path is Fig. 2 we can observe that $CH_4$ mixing ratio is the lowest above the mountains and slightly higher in the southern part which is flat. This shows that the $CH_4$ ratio has spatial dependency here which could be due to the low wind speeds and hot and convective conditions which leads in local temporal $CH_4$ accumulation and higher 'background' records in raster flights.

Line 181 – the term $CH_4$ tracers could be confusing, particularly those that associate tracers with the release of a different chemical compound from cylinders at a known rate to calculate CH4 (or other species) emissions from sources.

We agree that it can bring confusion with common use of tracer. Here we changed the "$CH_4$ tracer" to "model-based prognostic $CH_4$ tracers" with further explanation as follows in Sect. 2.3 (See L184-L192):

*To be able to geo-attribute emissions to certain emission clusters, we applied 33 individual model-based prognostic $CH_4$ tracers in the models which are transported according to the meteorological conditions. Each of these tracers represents the emissions of a specific area with a fixed emission rate of 1 g s$^{-1}$ or 3.6 kg hr$^{-1}$ and released at one individual or multiple release point(s). Meaning that one tracer represents the emissions of one or two clusters and one or two distant regions, assuming that they are sufficiently far away. This allows us to separate the signal of each cluster / region flown over or circled around. During the analysis, these tracers are not further considered, because, since the attribution by location is usually unambiguous.*

Line 201 – Unlike for the other inventories you do not specify which data the TNO-CAMS inventory provides.

*The TNO-CAMS v6.0 and EDGAR v7.0 inventories were both used to derive (partly) independent estimates of O&G and non-O&G $CH_4$ emissions in the target areas, and to calculate the share of O&G emissions in the total emissions in these areas (See L208-L210):*

*… **the Emissions Database for Global Atmospheric Research (EDGAR, 2023) v7.0 inventories were both used to calculate the percentage of O&G emissions to total emissions in the target areas.***

*The versions of the TNO-CAMS (v6.0) and EDGAR (v7.0) inventories used are now added in the manuscript. We extracted data from E-PRTR for year 2019, this is also added in the manuscript in 3.1.*

*… which includes a landfill listed in E-PRTR **for the year 2019,…***

Line 247 – it would be interesting to know why EDGAR has so few O&G emissions in these ROMEO regions.

*Agreed, it is interesting but beyond the scope of this investigation to determine why EDGAR v7.0 shows relatively low $CH_4$ emissions from O&G in these specific pixels. To do so, one would need access to the detailed data underlying calculations in EDGAR v7.0. Both inventories (TNO-CAMS v6.0 and EDGAR v7.0) calculate emissions at the national level because energy statistics are only available at that scale. It is likely that the spatial distribution proxy used by the EDGAR team does not accurately represent the production clusters in Romania. Apparently, the emissions in EDGAR v7.0 for the regions are mostly assigned to non-O&G emissions (see Table S7 in S5). .*

Line 258 – 'lowest value of each circle' – which instrument data is this referring to? Is the noise of the Aeris instrument baseline small enough that it does not result in a significant overestimate of peak height when dealing with peaks of 50-100 ppb?

*For clarification, we added '**retrieved from the Picarro instrument**' to the sentence (See L271).*

Line 349 – why do you need to show the confidence limits twice on the same line?

*We really don't need to have both. The sentence is now corrected as follows (See L350-L352):*

*Both EFs, 5.3 ± 2.0 kg hr$^{-1}$ site$^{-1}$ and 4.4 ± 1.7 kg hr$^{-1}$ site$^{-1}$, overlap with the EF of 5.4 kg hr$^{-1}$ (95% CI: 3.6 – 8.4 kg hr$^{-1}$) oil production site$^{-1}$ reported from ground-based measurements by Stavropoulou et al. (2023).*

Line 352 – 'from about dedicated measurements'?

*It is now corrected.*

Line 473 – '190 individual plumes evaluated'. Previously you say that 66 plumes were rejected due to upwind sources. At which stage of the evaluation were these rejected?

*For clarification we added and rephrased the lines in the manuscript as follows (See L494-L496):*

*A total of 256 plumes were identified, 66 of them were rejected, and 190 plumes were retained for analysis. Fig. 4 shows the plume area comparison of these 190 plumes from the SA mass balance flights and COSMO-GHG and MECO(3) models, respectively.*

Line 368 - 'possible underestimate of non-O&G emissions in the inventories for R7'. If you are comparing with inventory estimates at least give the emissions or refer the reader to S5 and which inventory you are using. As there is such a difference between inventories can you trust them to give a reliable estimate of the non-O&G sources. 3112 kg/hr for TNO is much closer to the flight estimates than 73 kg/hr from EDGAR. It seems that 5104 ± 1600 kg/hr (after upscaling to 100%) is within error of 7038 ± 1769 kg/hr.

*Indeed, the O&G emissions in the EDGAR inventory are significantly lower when compared to both the TNO-CAMS inventory and the measurements from the ROMEO campaign in Romania. However, the non-O&G emissions are comparable between the EDGAR v7.0 and TNO-CAMS v6.0 inventories. While we do not know the true scale of non-O&G emissions, we have chosen to use the absolute emissions values from the TNO-CAMS v6.0 inventory.*

*It is worth mentioning that it is not easy to draw this conclusion for the entire country or for one sector specifically. In the manuscript, we compare measurement-based emission quantifications with the inventories. However, generally speaking, we have found that EDGAR reports higher O&G related emissions for European countries compared to national reported data such as CAMS-REG and NIR reports (see Table 4 from Kuenen et al. (2022)). This is the case for countries like Norway and Netherlands. In these instances, the discrepancy arises from EDGAR using generic emission factors instead of lower country-specific emission factors. Therefore, we would expect a similar overestimation for Romania, so it is somewhat surprising that EDGAR reports lower emissions instead of equal or higher relative to other inventories. A comprehensive comparison is also hindered by the fact that spatial distribution plays an important role. It is possible that EDGAR locates the majority of emissions outside of the study domain. This aspect is beyond the scope of the study, as we focus on the region where we performed measurements.*

For clarification we added the following sentence to the manuscript (See L385-L393):

*While the measurement-based quantifications for region R7 from the two flights are 7,129 ± 2,097 kg hr$^{-1}$ and 6,947 ± 1,440 kg hr$^{-1}$, reported emissions for O&G activities in TNO-CAMS v6.0 and EDGAR v7.0 for this region were 3,112 kg hr$^{-1}$ and 73 kg hr$^{-1}$, respectively. This shows large difference between inventories and particularly a large underestimation in EDGAR v7.0 by a factor of about 100. The underestimation of O&G emissions from production areas in the earlier versions of EDGAR inventory has also been noted previously (Maasakkers et al., 2016; Scarpelli et al., 2020; Sheng et al., 2017). The causes and discrepancies of the difference observed between the measurements and the inventories require further investigation, which is beyond the scope of this study.*

And mentioned this underestimation in the abstract and conclusion as follows:

[in the abstract; See L47-L49] *We also observed large underestimation from O&G emissions in the Emissions Database for Global Atmospheric Research (EDGAR) v7.0 for our domain of study.*

[in the conclusion; See L608-L609] These results confirm that O&G methane emissions in 2019 were much higher than reported to UNFCCC *and estimated in EDGAR within our study domain.*

Line 381 – 'estimated emissions estimated at'.

Corrected.

Figure 4 – your dashed lines do not show up as dashes, even at 150% magnification.

Dashed lines in Figure 4 and Figure 5 are now adjusted.

Line 502 – Given that your calculated EF is 5.3 kg/hr per site (and the ground surveys we slightly higher), could you not have rerun the simulation with 1.5 g/s (5.4 kg/hr) to improve the comparison?

The transport of defined passive prognostic tracers is "linear," except for numerical limitations. Scaling the emission rate of such passive prognostic tracers will result in a likewise scaled mixing ratio, as the prognostic passive tracers are only subject to physical transport without any sink involved.

We have adopted the concept of "plume area," defined as the integration of methane enhancement along flight tracks, which is a function of the mixing ratio and flight path and is in unit of ppm * m. We calculated "plume areas" from measurements and model outputs to infer how the 1 g s$^{-1}$ fixed emission rate defined in the models differs from the average real-life O&G emission rate using the linear regression slope in Fig. 4 and Fig. 5.

Since the plume area is only a function of mixing ratio and flight track, and the flight tracks remain the same for different simulation settings, the "plume area" has a direct linear proportion to the increase in the fixed emission rate. This increase of the fixed 1 g s-1 emission rate in the model results in a change in the regression fit in Fig. 4 and Fig. 5, i.e. a shift of the measurements-models fits toward the 1:1 line.

Because the models are computationally expensive, we chose to utilize the approach presented in this paper and did not rerun the models to obtain a better fit.

**Supplementary:**

Fig S2 – There are farms in R5a and R8 regions. Did ground surveys detect these plumes and attempt to quantify them? Would be an alternative to a quite crude inventory, when attempting to subtract non-O&G emissions.

During the ground-surveys the ground-based teams did not focus on the farms and only targeted the O&G activities. Thus, we don't have detection and quantification from the ground-based surveys and could only rely on the numbers from inventories.

Table S6 – How can the 4 bottom-right cells (Sum R7 clusters and 100% fossil) be identical to Table 1 in the main paper, when they represent O&G in 2 very different inventories?

The absolute non-O&G emissions from the two inventories are only used when estimating O&G emissions for large regions. For smaller clusters, where we know there are no significant non-O&G sources, these emissions are not considered. The only exception is cluster R6C6, where a landfill was identified. This explains why the "Non-O&G emissions" column is left blank for the clusters in both Table 1 (main text) and Table S6 (Supplementary Information). Therefore, the four bottom-right cells you referenced are derived using the same values in both tables, assuming that 100% of the measured emissions are attributed to O&G activities, with no reliance on inventory data.

Table S7 – It is very concerning that there is such a big discrepancy in O&G emissions for the regions between the two inventories with TNO between 5 and 65 times higher than EDGAR. What is the difference in methodology that causes such difference?

This is indeed concerning, but on the other hand, it offers opportunities for prioritizing improvement. The sectors with the largest discrepancies should be the first to be further analyzed. In this manuscript, we did not investigate the possible causes of this difference in depth, as it would require detailed knowledge and access to the activity data and emission factors underlying EDGAR. This should be addressed in future studies by the inventory community. Moreover, these inventories are not built bottom-up, from individual activities at well sites, but at the national scale using national statistics in combination with emission factors, and then "down-scaled" using spatial proxy data. This means that the discrepancy could be from the calculated total emissions, but it could also be due to an issue with spatial allocation. Again, access to the proxy data underlying EDGAR would be needed for a deeper analysis. Although it is highly advisable to collaborate with the EDGAR team (not involved in the present study) to address this issue, it would be out of scope for the present study.

From the numbers in Table S7 we can see that the total emissions reported in EDGAR v7.0 are 60% of the cumulative sum of emissions in TNO-CAMS v6.0. The non-$CH_4$ emissions from the target region in Southern Romania in EDGAR v7.0 are actually about 40% higher than in TNO-CAMS v6.0. The O&G emissions reported in EDGAR v7.0 from the target region is only 3% of the reported O&G emissions from TNO-CAMS v6.0. In addition to different methodologies in building the inventories, it is possible that the methane emissions in EDGAR are assigned to wrong activities and/or the spatial proxy used is not a good representation of the O&G infrastructure in Romania. The latter is almost certainly true, as our study area is an important O&G region of Romania; even if underestimated the share of the region should be higher than currently represented in EDGAR v7.0.

**References**

Kuenen, J., Dellaert, S., Visschedijk, A., Jalkanen, J.-P., Super, I., and Denier van der Gon, H.: CAMS-REG-v4: a state-of-the-art high-resolution European emission inventory for air quality modelling, Earth Syst. Sci. Data, 14, 491–515, https://doi.org/10.5194/essd-14-491-2022, 2022.

Maasakkers, J. D., Jacob, D. J., Sulprizio, M. P., Turner, A. J., Weitz, M., Wirth, T., Hight, C., DeFigueiredo, M., Desai, M., Schmeltz, R., Hockstad, L., Bloom, A. A., Bowman, K. W., Jeong, S., and Fischer, M. L.: Gridded National Inventory of U.S. Methane Emissions, Environ. Sci. Technol., 50, 13123–13133, https://doi.org/10.1021/acs.est.6b02878, 2016.

Scarpelli, T. R., Jacob, D. J., Maasakkers, J. D., Sulprizio, M. P., Sheng, J.-X., Rose, K., Romeo, L., Worden, J. R., and Janssens-Maenhout, G.: A global gridded (0.1° × 0.1°) inventory of methane emissions from oil, gas, and coal exploitation based on national reports to the United Nations Framework Convention on Climate Change, Earth Syst. Sci. Data, 12, 563–575, https://doi.org/10.5194/essd-12-563-2020, 2020.

Sheng, J.-X., Jacob, D. J., Maasakkers, J. D., Sulprizio, M. P., Zavala-Araiza, D., and Hamburg, S. P.: A high-resolution (0.1° × 0.1°) inventory of methane emissions from Canadian and Mexican oil and gas systems, Atmos. Environ., 158, 211–215, https://doi.org/10.1016/j.atmosenv.2017.02.036, 2017.

Stavropoulou, F., Vinković, K., Kers, B., de Vries, M., van Heuven, S., Korbeń, P., Schmidt, M., Wietzel, J., Jagoda, P., Necki, J. M., Bartyzel, J., Maazallahi, H., Menoud, M., van der Veen, C., Walter, S., Tuzson, B., Ravelid, J., Morales, R. P., Emmenegger, L., Brunner, D., Steiner, M., Hensen, A., Velzeboer, I., van den Bulk, P., Denier van der Gon, H., Delre, A., Edjabou, M. E., Scheutz, C., Corbu, M., Iancu, S., Moaca, D., Scarlat, A., Tudor, A., Vizireanu, I., Calcan, A., Ardelean, M., Ghemulet, S., Pana, A., Constantinescu, A., Cusa, L., Nica, A., Baciu, C., Pop, C., Radovici, A., Mereuta, A., Stefanie, H., Dandocsi, A., Hermans, B., Schwietzke, S., Zavala-Araiza, D., Chen, H., and Röckmann, T.: High potential for CH4 emission mitigation from oil infrastructure in one of EU's major production regions, Atmos. Chem. Phys., 23, 10399–10412, https://doi.org/10.5194/acp-23-10399-2023, 2023.